# Early Diagnosis of Brain Diseases Using Artificial Intelligence and EV Molecular Data: A Proposed Noninvasive Repeated Diagnosis Approach

**DOI:** 10.3390/cells12010102

**Published:** 2022-12-26

**Authors:** Jae Hyun Park, Jisook Moon

**Affiliations:** 1TS Cell Bio, Wonju 26460, Republic of Korea; 2TS TECH, Wonju 26460, Republic of Korea; 3Department of Biotechnology, College of Life Science, CHA University, Seongnam 13488, Republic of Korea

**Keywords:** brain-derived extracellular vesicles, biomarker, diagnosis, machine learning

## Abstract

Brain-derived extracellular vesicles (BDEVs) are released from the central nervous system. Brain-related research and diagnostic techniques involving BDEVs have rapidly emerged as a means of diagnosing brain disorders because they are minimally invasive and enable repeatable measurements based on body fluids. However, EVs from various cells and organs are mixed in the blood, acting as potential obstacles for brain diagnostic systems using BDEVs. Therefore, it is important to screen appropriate brain EV markers to isolate BDEVs in blood. Here, we established a strategy for screening potential BDEV biomarkers. To collect various molecular data from the BDEVs, we propose that the sensitivity and specificity of the diagnostic system could be enhanced using machine learning and AI analysis. This BDEV-based diagnostic strategy could be used to diagnose various brain diseases and will help prevent disease through early diagnosis and early treatment.

## 1. Introduction

Extracellular vesicles (EVs) are secreted by almost all cells and are known to play an important role in intercellular communication [1]. EVs are classified into three subtypes: apoptotic bodies, microvesicles, and exosomes [2]. These EV subtypes are distinguished by their size and the characteristics of their biogenesis; however, specific markers are lacking for each EV subpopulation. Therefore, the overlap in size between subtypes ultimately leads to confusion over their identification. Given the lack of specific markers for each EV subtype and difficulty in distinguishing the subtypes, the International Society for Extracellular Vesicles has proposed the use of “EV” as a generic term for vesicles secreted from cells as well as the classification of subtypes based on EV size [3]. Exosomes, also called small EVs, are the smallest vesicles, measuring ~40–120 nm. They are produced through a complex process involving the internal budding of endosomes. Microvesicles, also called ectosomes, are ~100–1000 nm in size and are produced from the outer plasma membrane. Apoptotic bodies, the largest EVs, are ~500–4000 nm in size and are produced as a result of programmed cell death. Microvesicles are distinguished from apoptotic bodies by their size, and they differ by their formation, content, and membrane-specific antigens as microvesicles originate from the plasma membrane [4].

EVs carry biomolecules from origin cells and transmit them to other cells for cell communication. Attempts have been made to isolate tissue-specific EVs using their properties combined with tissue-specific single or multiple markers [5]. The successful isolation of EVs using tissue-specific markers enables the noninvasive and repeatable measurement of tissue pathology without tissue biopsy. These advantages have led to an increase in studies involving tissue-specific EV isolation and diagnosis.

Protocols to isolate brain-derived EVs by dissociation of brain tissue have been well established [6,7,8]. Papers that discover new disease mechanisms or biomarkers by analyzing the internal molecules in detail through cargo analysis in EVs have been reported. For example, Huang et. al. and Su et al. isolated EVs from the brains of AD patients and analyzed protein and lipid to find potential biomarkers [9,10].

In particular, considering that EVs can pass through the blood–brain barrier (BBB), there are also attempts to isolate EVs from blood for brain diagnosis. It is well known that amyloid-β (Aβ), tau, and α-synuclein, which are used as major biomarkers of brain disease, are released through EVs [11,12,13], and minimally invasive brain diagnosis can be made by detecting them in the blood. The brain is extremely difficult to biopsy, and in most cases, radiological methods are used for diagnosis, but the cost is high, and early diagnosis is not easy because the cause of the disease or the sensitivity is not high in the result. However, the minimally invasive brain diagnosis method using Blood EV not only reduces the physical and economic burden of the subject by enabling repetitive measurement, but also tracks individual changes by accumulating comparative data at different time points through repeated analysis. Therefore, changes in the brain can be continuously tracked, and diseases can be prevented through early diagnosis. In the first stage of diagnosis using BDEV, if combined with existing diagnostic methods, it will contribute to early diagnosis, and it is expected that personalized diagnosis and prevention will be possible if data are accumulated in the second stage further through repetitive measurement. Recently, in order to further increase the accuracy of body fluid-based diagnosis, reports on the results of separating BDEV from blood and using it for clinical diagnosis have been steadily made. In the case of traumatic brain injury, there was no appropriate diagnostic molecular marker, but by examining miRNA and protein profiles of blood EVs isolated using neuronal-specific markers, it was shown that there is a possibility to distinguish between patients and controls [14,15,16,17], and Alzheimer’s disease. Regarding disease, isolated EVs were tested for traditional AD markers such as tau and Aβ, showing the possibility of predicting AD in the early stages of the preclinical course [18,19]. In addition, astrocyte-derived EV has higher expression levels of proteins known to be associated with AD pathogenesis, such as amyloid beta precursor protein (APP), γ-secretase, and β-secretase, than neuron-derived EV, so it is likely to be more useful for diagnosing AD patients [20]. Although α-synuclein is used as an important biomarker for PD, it is known that the initial false positive rate is high. When α-synuclein of neuron-derived EV is investigated, a more reliable relationship between PD and healthy control is seen than when using total plasma [21].

Furthermore, it has been reported that by isolating oligodendrocyte-derived EVs and examining α-synuclein, multiple system atrophy and PD, which are often clinically confused, can be more accurately distinguished [22,23]. In the case of microglia exosomes, many share biomarkers with macrophages and monocytes, making it difficult to separate them specifically [24]. However, after measuring microglia-derived EV in the blood using plant lectin, isolectin B4, which is known to bind to microglia, and CD11b, a microglia surface marker, it was reported that the amount increased in AD patients [25].

In the field of disease diagnosis, researchers have attempted to indirectly assess the state of brain tissue using the following process.

(1) The isolation of EVs from the blood; (2) the enrichment of brain-derived EVs (BDEVs) using an immunocapture method with brain-related biomarkers; and (3) analysis of the cargoes of BDEVs.

EVs secreted from the brain pass through the BBB and are found in the blood [5]; thus, to improve the diagnosis of brain disorders, it is important to enrich BDEVs effectively by separating them from various organ-derived EVs.

In the present study, a strategy for identifying BDEV biomarkers to improve the enrichment of BDEVs and a method for applying this strategy to brain diagnosis and brain region-specific discovery were proposed and reviewed. Our suggested strategy is summarized with three steps: (1) considerations of new BDEV biomarker discovery, (2) application of BDEV, and (3) machine learning and AI strategies for clinical application. Through the discovery of more effective BDEV markers, we hope to be able to achieve the early diagnosis of brain diseases.

## 2. Results

### 2.1. Concept of BDEVs

BDEVs are EVs secreted from various cells in the brain. They are known to pass through the BBB and are secreted into the blood. BDEVs contain various molecules, including noncoding RNAs, mRNAs, proteins, and metabolites, that reflect their origin cell [26]. Thus, by capturing BDEVs in the blood and examining their internal molecules, the state of the brain can be identified indirectly and disease can be predicted (Figure 1) [20,22,23,27,28,29,30]. For example, L1 neuronal cell adhesion molecule (L1CAM) was considered a BDEV marker, and attempts were made to analyze brain diseases via L1CAM-positive EVs in the blood [19,21,31]. However, some studies have shown that L1CAM has low brain specificity and is expressed in cells other than brain cells; hence, the role of L1CAM as a BDEV is controversial. Therefore, more specific biomarkers must be identified to improve the accuracy of brain disease diagnosis.

### 2.2. Proposed BDEV Biomarker Discovery Strategy

To discover novel BDEV biomarkers for blood biopsy, it is necessary to sequentially satisfy at least the following four conditions: (1) specific expression in brain tissue, (2) known molecules present in EVs (3) present in EV membranes for BDEV enrichment via immune capture from EVs of various origins, and (4) known biomarkers previously detected in blood (Figure 2).

#### Four Steps for Discovering BDEV Biomarkers

1.Brain-specific biomarker

One weakness of existing BDEV biomarkers is that they are also expressed in other tissues. Therefore, it is essential that more brain-specific biomarkers are discovered. Given that the National Center for Biotechnology Information (NCBI) and other databases provide human tissue RNA expression data, it is possible to specify genes expressed only in the brain using this data. In addition, The Human Protein Atlas provides immunohistochemistry results for all organs; therefore, it is also possible to confirm protein expression levels.

2.Known to have been detected in EVs

Regardless of brain-specific expression, a gene cannot be used in a BDEV-based strategy if it is not loaded into an EV. Therefore, it is necessary to select a biomarker that has been detected in EVs or is likely to exist in EVs. As the demand for information on EVs has increased, databases such as ExoCarta and Vesiclepedia have begun to provide data on EV cargoes [32,33]. In addition to providing gene annotation information, the Gene Ontology (GO) resource provides information on the location of a gene product in the cell [34]; thus, annotation information on the gene products that exist in EVs is available. As research on EVs has progressed, our understanding of whether proteins are loaded into EVs has also improved. Two EV cargo loading mechanisms exist: (1) proteins are loaded into EVs via the endosomal system (exosome) and secreted and (2) proteins are loaded together by membrane shedding (microparticles) [4]. Based on these basic biological processes, we can infer whether endosomal system proteins or membrane proteins are likely to be loaded into EVs. Some studies have shown that certain protein motifs facilitate EV loading [35]. Improving our understanding of the motifs that promote EV loading will improve our ability to predict the biomarkers likely to be loaded into EVs. If databases and the available information are used effectively, it will become possible to select a specific protein biomarker known to be EV cargo or tentatively identify cargo candidates with a high probability of success.

3.Isolation of BDEVs using various methods

EVs derived from various organs are mixed in the blood, and it is important that only BDEVs are enriched for brain diagnosis. The methods most frequently used for such enrichment are antigen–antibody affinity methods, such as immunoprecipitation. Therefore, any biomarker selected for BDEV enrichment should possess high accessibility to the antibody. It is difficult for an antibody to penetrate the EV, which is protected by a lipid membrane; thus, the BDEV biomarker should exist outside the EV in the same form as a membrane protein. Indeed, most of the proteins currently used as cell- and tissue-specific EV biomarkers are classified as membrane proteins and receptors according to the GO term “cellular component” classification. A technology for labeling the internal RNA cargo of EVs through EV fusion has been studied [36]. By applying this method, not only tissue-specific proteins but also non-coding RNAs such as miRNA/lncRNA, etc., with known tissue or cell specificity, can be sorted through appropriate labeling. In addition to the separation method using antigen-antibody affinity, it is also possible to capture only specific EVs simply by using ligand-receptor interaction. If other such methods are developed, the range of choices for BDEV biomarkers would be widened.

4.BDEV markers detected in the blood: an optional step for isolating BDEVs

Even when a putative biomarker is identified using the abovementioned criteria, the existence of the putative biomarker with the EV and whether it can pass through the BBB into the blood are possible issues because transport routes remain obscure [37]. To minimize errors related to these problems, it is necessary to at least guarantee that the biomarker has been detected in the blood. The Human Protein Atlas provides information on the level of protein expression in the blood in connection with the PeptideAtlas database, and the EV cargo database ExoCarta provides the origin of EVs, i.e., information on blood EVs, cell culture EVs, etc. EV-related data are now being submitted to proteomics databases, e.g., ProteomeXchange; therefore, it is possible to determine whether a biomarker has been detected in the blood using thesis or public proteomics data from a database. The negative effects of trial and error can be minimized by using databases to obtain the minimum required data prior to beginning the research.

### 2.3. Strategy of Brain Region- or Cell Type-Specific Diagnosis Using Multiple Markers

The human brain has brain regions with specialized functions and different cell subtype compositions; thus, the molecular and biological characteristics of brain diseases that occur in each brain region are also different, and the treatment for these diseases differs accordingly. Therefore, discovering brain region-specific markers in the brain is important for diagnosing different brain diseases (Figure 3). It is difficult to accurately diagnose a specific brain disease in its early stage because the early pathological characteristics of brain diseases are often similar and mixed [38,39]. Hence, using BDEVs for brain region- or cell type-specific diagnosis at the molecular level could make it possible to identify brain diseases more accurately at an earlier stage. The development of analysis technology and the accumulation of data have led to the identification of brain region- and cell type-specific expression biomarkers. The Human Protein Atlas provides RNA expression information for each brain region in humans, and the Allen Brain Atlas also provides this information or in situ hybridization results. In particular, single cell analysis has led to the discovery of brain cell type-specific biomarkers [40,41]. Establishing a biomarker capable of isolating bulk BDEVs secreted from the entire brain will facilitate a fuller analysis by enabling the double-capture of additional region- or cell type-specific biomarkers, even if these biomarkers are not brain-specific. For example, EV secretion is known to be increased in cancer cells [42]; if EV secretion changes according to the disease state, such as in cancer, assessing the subpopulation of each BDEV biomarker will help accurately diagnose the disease [43]. Alternatively, new diagnosis systems could be constructed using molecular monitoring of brain-derived region- or cell type-specific EVs.

### 2.4. Repeatable and Minimally Invasive Brain Monitoring System

By analyzing blood-based BDEVs, we can indirectly identify the state of the brain using a minimally invasive method that is less burdensome than repeated biopsies. Moreover, using this method, it will be possible to build a BDEV-based sustainable brain monitoring system (Figure 4). Such a system could capture the prognosis of a disease through the periodic collection of the patient’s blood, isolation of BDEVs, and continuous monitoring of disease-related molecules. The measured molecule may be a well-known biomarker (e.g., amyloid beta, amyloid precursor protein, tau, synuclein, and neuron-specific enolase) or a new biomarker discovered using the above-described BDEV-based brain disease diagnosis strategy. Furthermore, if information from various research fields is accumulated, it may be possible to build an AI-based automated diagnosis system through machine learning.

### 2.5. Machine Learning, AI, and Personal Diagnosis

The development of high-throughput data production methods, e.g., sequencing and mass spectrometry, has facilitated the collection of molecular data from individual patients. By combining this data with machine learning, AI-based diagnosis is becoming possible. The biological activities of humans are controlled by complex interactions at various levels such as the transcriptome, proteome, and metabolome. In addition, a symbiotic microbiome within humans is being actively studied in an attempt to explain cellular activities more accurately (Figure 5). Rather than attempting to understand biological activities in only one major field, which was once the standard, the era of multiomics has enabled researchers to study biological activities in several different biological layers [44]. Given the complexity of these interactions, machine learning, a field of AI in which data are interpreted via computational methods and universal patterns are derived from the data, is now being applied to help interpret multiomics data, which is difficult to interpret using traditional human-led methods. Moreover, studies have been conducted to apply the results of machine learning to disease diagnosis.

Feature selection is the process by which data are selected for machine learning (Table 1). It is an important step for reducing data noise and increasing computational power. Humans are known to possess about 20,000–25,000 genes; if the RNA and protein expression levels of all human genes are used for machine learning, the number of data combinations increases exponentially and the computational burden becomes overwhelming. Furthermore, as the number of features used in machine learning increases, the model becomes more complex and difficult to interpret. Therefore, it is important to select only the appropriate features for obtaining the desired results, and the selected features themselves can be applied as new biomarkers. Features can be selected by comparing those with and without disease using existing statistical methods, although various tools have been developed specifically for feature selection [45,46].

After discovering biomarkers using appropriate feature selection applied to complex molecular data, the biomarkers must then be used for disease diagnosis. Hence, various algorithms have been developed for making clinical decisions. Machine learning is applied after collecting molecular data for biopsy, such as data from plasma and urine measurements, but it is difficult to account for the tissue specificity of the target disease. However, using a BDEV platform to indirectly obtain tissue-specific molecular data that are difficult to obtain via biopsy and applying these data using a machine learning approach will improve brain disease diagnosis. In addition, collecting and recording patient data over an extended period will help predict patient prognosis.

Table 2 summarizes studies to diagnose brain diseases using EVs isolated from blood or brain-derived EVs and machine learning. After obtaining RNA or protein expression using sequencing and mass spectrometry, in some cases, features selected through a feature selection step are used, and clinical features related to patients such as sex, age, and behavioral score, etc., and there are also studies using machine learning by quantifying known brain disease-related proteins such as aβ, α-synuclein, and tau, etc. by immunocapture assay. These studies show that EV-based diagnosis strategies can not only discover new disease features but also develop known molecular features.

## 3. Conclusions

A BDEV-based brain diagnosis platform could be used to indirectly obtain molecular data from brain tissue that is otherwise difficult to obtain via biopsy. Additionally, the identification and selection of appropriate BDEV biomarkers will facilitate the analysis of various brain region- or cell type-specific BDEVs and the accurate diagnosis of related diseases. In addition, the molecular data produced by analyzing BDEVs combined with machine learning tools could help explain various brain phenomena and will help discover new brain disease biomarkers. Ultimately, the collection of patient data over an extended period will enable BDEV-based disease diagnosis and improve our predictions of patient prognosis.

## Figures and Tables

**Figure 1 cells-12-00102-f001:**
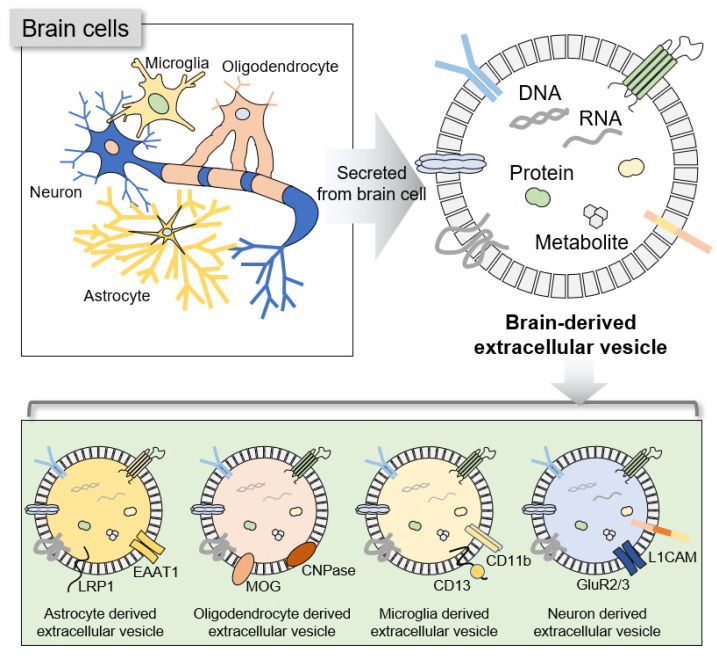
Brain-derived extracellular vesicles. Various cell types are found in the brain, including neurons, oligodendrocytes, astrocytes, and microglia, and these cells secrete extracellular vesicles (EVs) that reflect the characteristics of their origin cells. These EVs carry various cargoes such as DNA, RNA, proteins, and metabolites. Given that EVs share the characteristics of their origin cell, it is possible to sort cell type-specific EVs using the surface markers of each cell type in the brain.

**Figure 2 cells-12-00102-f002:**
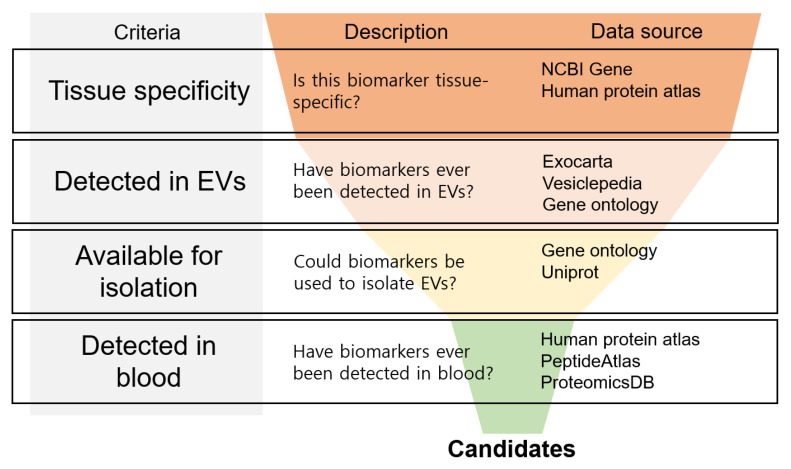
Selection strategy for novel BDEV markers. The step-by-step criteria for identifying novel BDEV biomarker candidates are shown. First, the marker should have brain tissue specificity. Second, the marker should be highly likely to be or have been detected in an EV. Third, the biomarker should be sortable as a BDEV in a mixture of EVs. Finally, the biomarker should ideally have been detected in the blood.

**Figure 3 cells-12-00102-f003:**
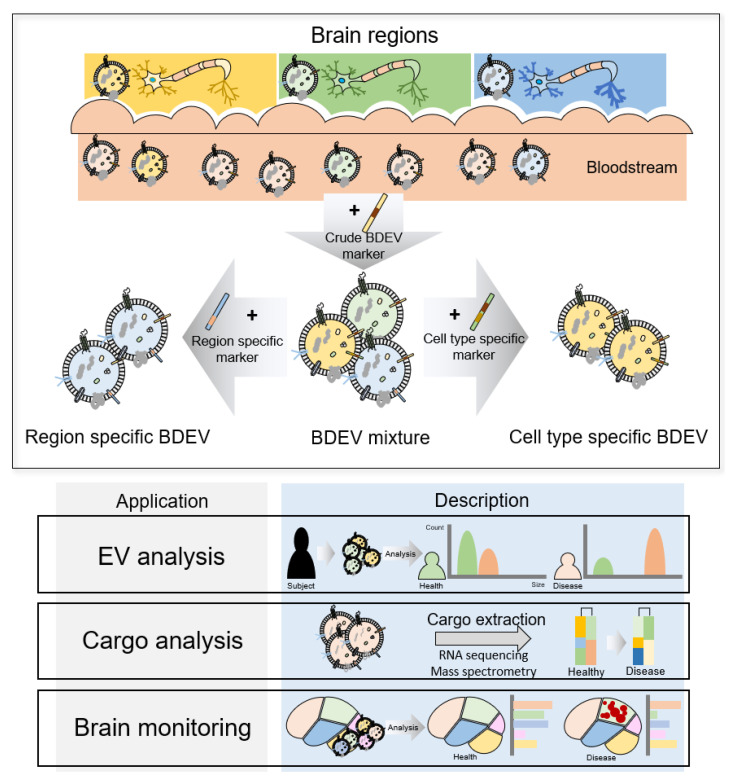
Application of brain region-specific BDEVs. Brain cells are differentiated into cells with specialized functions depending on the brain region, even if the cells are of the same lineage, e.g., cholinergic neurons in the hippocampus and dopaminergic neurons in the substantia nigra.

**Figure 4 cells-12-00102-f004:**
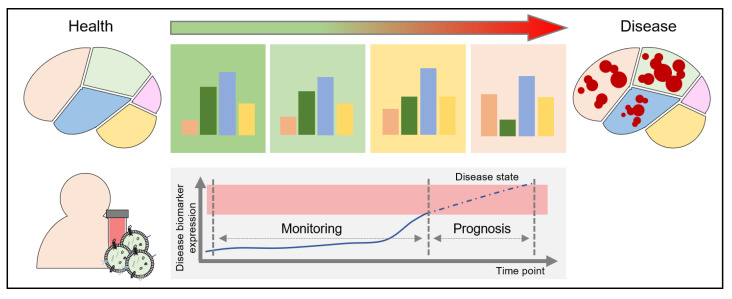
Repeated measurement of BDEVs. BDEVs can serve as a tool for diagnosing brain diseases, including those related to blood-based brain aging. As the state of the body is monitored through periodic blood tests, BDEVs in the blood can be periodically separated and measured through these tests. In addition, it is possible to continuously trace the change in the pattern of the internal disease-associated biomarker; therefore, it is possible to establish a blood-based, brain-specific, long-term, and repeatable testing method by comparing and analyzing not only age-related changes but also changes in groups of participants with a similar age.

**Figure 5 cells-12-00102-f005:**
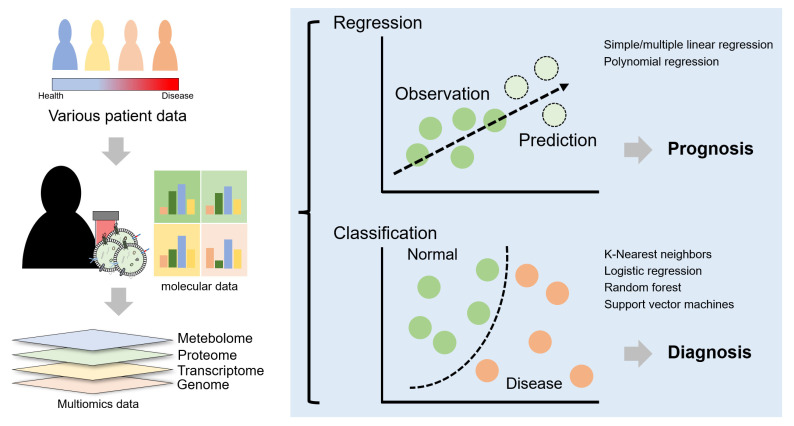
BDEVs and machine learning. Various multiomics data can be collected through molecular data analyses of BDEV cargoes, and the efficiency of disease prediction and diagnosis can be improved using machine learning. This process will enable not only personalized diagnosis but also personalized medicine.

**Table 1 cells-12-00102-t001:** An example of feature selection methods.

Method		Description	
Filter	Statistical techniques for evaluating the feature score/rank	*t*-test/ANOVA	Traditionally used to compare two or more categorical groups and continuous features
		Chi-square test	Used for comparisons of categorical features
		Pearson’s correlation	Measure of the linear relationship and dependence between two or more features
		Informatics gain	Measure of the best features providing optimal information about a class
Wrapper	Select the best feature subset based on classification performance	Forward selection	Begins with zero features and iteratively adds features that provide predictive power
		Backward elimination	Begins with all features and iteratively removes features that do not provide predictive power
		Stepwise selection	Combination of forward selection and backward elimination
Embedded	Search method for obtaining the best subset of features that is built into a learning algorithm	Ridge/LASSO regression	Type of linear regression that uses feature penalty
		Decision tree/random forest	Classifier that combines many decision trees

**Table 2 cells-12-00102-t002:** Machine learning studies with blood extracellular vesicle for the diagnosis of brain diseases.

Sample	Target Disease	Input Data	Machine Leaning Algorithm	Reference
Plasma EV	Alzheimer’s disease	miRNA sequencing	Decision trees, Support vector machine, adaBoost	[47]
L1CAM+ plasma EV	Alzheimer’s disease	Clinical features, Tau, Aβ, IRS-1, EV concentration and diameter	Logistic regression	[48]
L1CAM+ plasma EV	Cognitive deficits in HIV	Clinical features, HMGB1, Neurofilament light, Tau	K-Nearest Neighbor, Support vector machine, adaBoost	[49]
Plasma EV	Parkinson’s disease	Clinical features, Tau, Aβ, α-synuclein	Artificial neural network	[50]
Plasma EV	Parkinson’s disease and multiple system atrophy	Surface antigen	Random forest	[51,52]
GluR2+ plasma EV	Traumatic brain injury	miRNA sequencing	K-Nearest Neighbor, Support vector machine, Linear discriminant analysis, Logistic regression, Naive Bayes	[16]
Plasma EV	Traumatic brain injury	Proteomics	Ensemble Learning	[53]

## Data Availability

Not applicable.

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
