# Peer review of "Early Diagnosis of Brain Diseases Using Artificial Intelligence and EV Molecular Data: A Proposed Noninvasive Repeated Diagnosis Approach"

_cells, 2022, doi:10.3390/cells12010102_

Round 1
Reviewer 1 Report
This manuscript is actually not so much a commentary as an hypothesis, which however is nicely discussed. What it totally lacks is references to the most recent published progress in BDEVs which provide practical examples of biomarkers that could be validated. The authors should consider citing and discussing at least these papers, the majority of which also discuss applications of machine learning:
# Bergauer et al., Biomed Pharmacother. 2022 Feb;146:112602. doi: 10.1016/j.biopha.2021.112602
# Brenna et al., J Extracell Vesicles 2020 Aug 27;9(1):1809065. doi: 10.1080/20013078.2020.1809065
# Beard et al., Brain Commun 2021 Jul 8;3(3):fcab151. doi: 10.1093/braincomms/fcab151
# Ko et al., J Neurotrauma. 2020 Nov 15;37(22):2424-2434. doi: 10.1089/neu.2018.6220
# Ko et al., Lab Chip. 2018 Dec 7;18(23):3617-3630. doi: 10.1039/c8lc00672e
Frontiers had an entire Research Topic on this ("An Approach of Brain Derived Extracellular Vesicles in Diagnosis and Prognosis of Brain Pathologies"); see https://www.frontiersin.org/research-topics/27519/an-approach-of-brain-derived-extracellular-vesicles-in-diagnosis-and-prognosis-of-brain-pathologies#articles
Without such discussions it is difficult to see what value this Commentary would add to the body of BDEV research.
In addition, the following edits have to be made.
# Introduction, lines 27 & 28: The first half of the second sentence is a direct duplication of the first.
# p. 3, line 77: Define L1CAM as the L1 neuronal cell adhesion molecule
# p. 4, lines 142-144 runs: "Even when a putative biomarker is identified using the above-mentioned criteria, the existence of the putative biomarker with the EV and whether it can pass through the BBB into the blood are possible issues. Because mechanisms of transport from the 144 brain to the blood is still unclear [23]." These two sentences should be merged.
Author Response
We really appreciate your great comments that help us to improve our research and manuscript. We sincerely respond all issues reviewers suggested as much as we can and addressed your suggestions in our revised manuscript.
This manuscript is actually not so much a commentary as an hypothesis, which however is nicely discussed. What it totally lacks is references to the most recent published progress in BDEVs which provide practical examples of biomarkers that could be validated. The authors should consider citing and discussing at least these papers, the majority of which also discuss applications of machine learning:
# Bergauer et al., Biomed Pharmacother. 2022 Feb;146:112602. doi: 10.1016/j.biopha.2021.112602
# Brenna et al., J Extracell Vesicles 2020 Aug 27;9(1):1809065. doi: 10.1080/20013078.2020.1809065
# Beard et al., Brain Commun 2021 Jul 8;3(3):fcab151. doi: 10.1093/braincomms/fcab151
# Ko et al., J Neurotrauma. 2020 Nov 15;37(22):2424-2434. doi: 10.1089/neu.2018.6220
# Ko et al., Lab Chip. 2018 Dec 7;18(23):3617-3630. doi: 10.1039/c8lc00672e
- Thank you for your good suggestions. In order to solidify the discussion on BDEV, we added several references to the introduction, including previous and recent BDEV-related studies and papers you recommended. Also, we cited the papers discussing applications of machine learning.
Without such discussions it is difficult to see what value this Commentary would add to the body of BDEV research.
- We really appreciate your comments. We totally agree with the point. As the first reviewer mentioned, we added explanation in Introduction. Most changes are in red in the manuscript. (page 2)
In addition, the following edits have to be made.
# Introduction, lines 27 & 28: The first half of the second sentence is a direct duplication of the first.
# p. 3, line 77: Define L1CAM as the L1 neuronal cell adhesion molecule
- Thank you so much for your careful check and we have changed as the first reviewer suggested (page 1 and page 3)
# p. 4, lines 142-144 runs: "Even when a putative biomarker is identified using the above-mentioned criteria, the existence of the putative biomarker with the EV and whether it can pass through the BBB into the blood are possible issues. Because mechanisms of transport from the 144 brain to the blood is still unclear [23]." These two sentences should be merged.
- Again, thank you for all of your comment and have merged two sentences on page 5 as follows:
- Even when a putative biomarker is identified using the abovementioned criteria, the existence of the putative biomarker with the EV and whether it can pass through the BBB into the blood are possible issues because transport routes remain obscure.

Reviewer 2 Report
This paper proposed to use AI and machine learning for early diagnosis of brain diseases. BDEVs are released from the central nervous system and they can be used to diagnose brain diseases with minimally invasive and it enables repeatable measurements based on body fluids. However, the EV subtypes are hard to distinguish by size and the characteristics using existing database. The authors proposed a BDEV biomarker discovery strategy with four steps, and use machine learning techniques such as features selection. The approach can detect brain diseases more accurately at an earlier stage.
In general the topic is very interesting to readers, and the proposed approach has great impact on early detection of brain diseases based on body fluids. The major concern is the machine learning algorithm and experiment part is missing. Only table 1 listed some feature selection methods, in which most are statistic approaches not machine learning. AI and machine learning can automatically select useful features despite the huge size of gene. In image and video processing, there could be millions of features, so the complexity of RNA and protein expression is not a big problem. One problem maybe the labeled data is hard or expensive to obtain for BDEV recognition. There are some useful references [1] for big data assimilation, and [2] for sequential data processing.
[1] Distributed mean-field-type filters for Big Data assimilation, in the second IEEE International Conference on Data Science and Systems (HPCC-SmartCity-DSS), Sydney, Australia, Dec, 2016, pp. 1446-1453.
[2] Correlative Mean-Field Filter for Sequential and Spatial Data Processing, in the Proceedings of IEEE International Conference on Computer as a Tool (EUROCON), Ohrid, Macedonia, July 2017
Author Response
We really appreciate your great comments that help us to improve our research and manuscript. We sincerely respond all issues reviewers suggested as much as we can and addressed your suggestions in our revised manuscript.
This paper proposed to use AI and machine learning for early diagnosis of brain diseases. BDEVs are released from the central nervous system and they can be used to diagnose brain diseases with minimally invasive and it enables repeatable measurements based on body fluids. However, the EV subtypes are hard to distinguish by size and the characteristicsusing existing database. The authors proposed a BDEV biomarker discovery strategy with four steps, and use machine learning techniques such as features selection. The approach can detect brain diseases more accurately at an earlier stage.
- We really appreciate Reviewer #2’s careful comments and agree with that EV subtypes such as exosome, MV, etc are not easy to distinguish using our proposed method. Because it is difficult to compare not only DB but also technically, we tried to classify EVs according to cell/organ origin.
In general the topic is very interesting to readers, and the proposed approach has great impact on early detection of brain diseases based on body fluids. The major concern is the machine learning algorithm and experiment part is missing. Only table 1 listed some feature selection methods, in which most are statistic approaches not machine learning. AI and machine learning can automatically select useful features despite the huge size of gene. In image and video processing, there could be millions of features, so the complexity of RNA and protein expression is not a big problem. One problem maybe the labeled data is hard or expensive to obtain for BDEV recognition. There are some useful references [1] for big data assimilation, and [2] for sequential data processing.
[1] Distributed mean-field-type filters for Big Data assimilation, in the second IEEE International Conference on Data Science and Systems (HPCC-SmartCity-DSS), Sydney, Australia, Dec, 2016, pp. 1446-1453.
[2] Correlative Mean-Field Filter for Sequential and Spatial Data Processing, in the Proceedings of IEEE International Conference on Computer as a Tool (EUROCON), Ohrid, Macedonia, July 2017
- Thanks again for your precious comment. The method of obtaining labeled data is a big challenge not only in the field of diagnosis but also in all AI fields. Fortunately, however, it is known that EVs exist relatively and stably in biological fluids and that EVs can be isolated from them. Thanks to these characteristics, it is expected that after separating BDEV from frozen samples stored by research institutes without new sample collection, it will be possible to produce molecular data from it and use it in a diagnosis system using AI.
- Also, based on the articles you recommended, we supplemented ML and experimental parts that were lacking in our manuscripts. All changes are in red on pages 9~10.

Reviewer 3 Report
Overview and general recommendation:
This manuscript developed a BDEV-based brain diagnosis platform for screening potential BDEV biomarkers that are beneficial to accurate diagnosis of related brain diseases by using machine learning tools. Generally, the manuscript is well-written, except for the introduction section and some errors in the reference section. Even though the research is novel, it would be more appropriate to publish this work in other journals instead of Cells because Cells aims at experimental but not theoretical data. Some comments have been shown below.
Major Comments:
1.The purpose of this study can be more concise in the Abstract section.
2. Please exemplify some research about strategies to assess the state of brain tissue and cite some references.
3. The last graph of the introduction section should depict this work in detail.
4. What are the current methods for collecting BDEVs? Compared to these methods, what is the advantage of this approach? These advantages should be mentioned.
5. What is the purpose of selecting brain diseases as an object? It is important to describe in the abstract section.
6. The established BDEV-based brain diagnosis platform needs to be applied to some samples to show the currency and practicality compared with biopsy.
7. It seems that the BDEV-based brain diagnosis platform did not offer any significant results, which was not convincing.
Minor Comments:
1. Please keep the format of all references consistent in the Reference section. The pages of reference two and six were missing. Please read through the manuscript and revise it.
Author Response
We really appreciate your great comments that help us to improve our research and manuscript. We sincerely respond all issues reviewers suggested as much as we can and addressed your suggestions in our revised manuscript.
This manuscript developed a BDEV-based brain diagnosis platform for screening potential BDEV biomarkers that are beneficial to accurate diagnosis of related brain diseases by using machine learning tools. Generally, the manuscript is well-written, except for the introduction section and some errors in the reference section. Even though the research is novel, it would be more appropriate to publish this work in other journals instead of Cellsbecause Cells aims at experimental but not theoretical data. Some comments have been shown below.
- We put a lot of thought into answering your comment that our manuscript is not suitable for Cells. It may seem that the content of our manuscript does not fit the scope of Cells. Our Lab has continued to conduct EV research from a therapeutic and diagnostic perspective. We are discovering markers that show the possibility that EVs can be used for brain diagnosis and actively conducting experimental verification.
- In the hope that the process of discovering biomarkers capable of isolating brain-derived EVs from blood or body fluids will be universally known, this paper was submitted to the commentary section of Cells rather than the original article.
Major Comments:
The purpose of this study can be more concise in the Abstract section.
- Thank you for your comment. We deleted some unnecessary and unclear statements of abstract and modified them for more clarity. The correction is on page 1 as follows:
- Brain-derived extracellular vesicles (BDEVs) are released from the central nervous system. Brain-related research and diagnostic techniques involving BDEVs have rapidly emerged as a means of diagnosing brain disorders because they are minimally invasive and enable repeatable measurements based on body fluids. However, EVs from various cells and organs are mixed in the blood, acting as potential obstacles for brain diagnostic systems using BDEVs. Therefore, it is important to screen appropriate brain EV markers to isolate BDEVs in blood. Here, we established a strategy for screening potential BDEV biomarkers. To collect various molecular data from the BDEVs, we propose that the sensitivity and specificity of the diagnostic system could be enhanced using machine learning and AI analysis. This BDEV-based diagnostic strategy could be used to diagnose various brain diseases and will help prevent disease through early diagnosis and early treatment.
Please exemplify some research about strategies to assess the state of brain tissue and cite some references.
- Reviewer # 1 gave the same comment as reviewer # 2 to add more discussion of BDEVs. We added a paragraph on the example of diagnosis using BDEV to the introduction and added references. Thank you very much for improving our manuscript through your comments (page 2).
The last graph of the introduction section should depict this work in detail.
- Thank you for your As you suggested, a brief description of this manuscript has been added in the last paragraph of the introduction (page 3).
What are the current methods for collecting BDEVs? Compared to these methods, what is the advantage of this approach? These advantages should be mentioned.
- Currently, the method mainly used for BDEV collection is immunoprecipitation. However, in some papers, in addition to antigen-antibody interaction, there was also a method of detecting only specific EVs by labeling internal RNA without ligand-receptor interaction using peptides or EV lysis.
- These methods have the advantage of being able to isolate EVs using more diverse targets, but direct comparison is difficult as there are still few related studies. If it can be separated by the surface membrane marker of EV, it has the advantage of being easily detected by antigen-antibody reaction. In addition, it has the advantage of being able to perform cargo analysis within the EV to secure more accurate diagnostic specificity and sensitivity using the primary separated BDEV. A related description was added to the manuscript (page5).
What is the purpose of selecting brain diseases as an object? It is important to describe in the abstract section.
- To the best of our knowledge, there is no method to measure brain disease diagnosis repeatedly and noninvasively. This is because biopsy is almost impossible due to the nature of the organ, and it relies on expensive examination methods such as CT and MRI. Therefore, we paid attention to EVs as a way to examine the brain in a relatively simple way. I will add this part to the abstract and introduction. (page2).
The established BDEV-based brain diagnosis platform needs to be applied to some samples to show the currency and practicality compared with biopsy.
- Conventionally, the L1CAM marker has been mainly used to isolate BDEV. Papers using L1CAM have also been published, but in the case of L1CAM, there is a controversial issue mainly due to poor brain specificity. Since there is no newly discovered BDEV marker after the L1CAM marker, this study proposes a method for discovering a new BDEV marker. We hope that the new discovery strategy proposed in this manuscript will be helpful for the establishment of more accurate BDEV biomarkers. In addition, we are also actively conducting verification experiments using novel BDEV markers discovered through this method. As biomarkers with higher specificity are discovered and EV isolation and purification technologies are advanced, the developing BDEV platform, which is more accessible than biopsy, would become competitive.
It seems that the BDEV-based brain diagnosis platform did not offer any significant results, which was not convincing.
- Thank you for your thoughtful and candid comments. Currently, it appears that breakthroughs have not been made due to confusion with issues related to EV purification and characterization. Nevertheless, research papers for discovering new biomarkers for brain diseases and establishing diagnostic strategies using BDEV at the clinical level appeared one after another, and related papers were added to the introduction of the current manuscript. We believe that precise and more brain-specific BDEV biomarker discovery will address these issues. In addition, although it will not completely replace the existing brain diagnosis method in the first stage of using BDEV, the BDEV-based diagnosis platform will show more accurate diagnostic tools as data from isolated BDEVs accumulate, which will be the next step in AI or machine learning. will be helpful and required.
Minor Comments:
Please keep the format of all references consistent in the Reference section. The pages of reference two and six were missing. Please read through the manuscript and revise it.
- Thank you for your comment. We have checked the format of all references, fixed mistakes, and make them consistent. Thanks again.

Round 2
Reviewer 1 Report
Now fine as a commentary/hypothesis.
These two sentences should be merged.: "Even when a putative biomarker is identified using the abovementioned criteria, the existence of the putative biomarker with the EV and whether it can pass through the BBB into the blood are possible issues. Because mechanisms of transport from the 144 brain to the blood is still unclear [37]."
Author Response
These two sentences should be merged.: "Even when a putative biomarker is identified using the abovementioned criteria, the existence of the putative biomarker with the EV and whether it can pass through the BBB into the blood are possible issues. Because mechanisms of transport from the 144 brain to the blood is still unclear [37]."
- Thanks for checking. In fact, we corrected this in the last revision, but accidentally deleted it. In the current manuscript, the following modifications were made: Even when a putative biomarker is identified using the abovementioned criteria, the existence of the putative biomarker with the EV and whether it can pass through the BBB into the blood are possible issues because transport routes remain obscure.

Reviewer 3 Report
Overview and general recommendation:
This manuscript has been improved after revision based on the comments from reviewers. However, there are still some minor mistakes in the manuscript. The authors need to read through the manuscript to correct these errors before publication. Some comments have been shown below.
Minor Comments:
1. In the abstract section, the font size of ‘To collect various molecular data from the’ obviously is different from other words.
2. In line 51, page 2: “In particular, considering that EVs can pass through the BBB, there are also attempt…”. BBB should not be abbreviated since it appeared in the manuscript for the first time.
3. In line 179, page 5: “…RNAs such as miRNA/lncRNA etc.” There should be comma in front of ‘etc’.
4. In lines 285-292, page 9, please make sure that all sizes and fonts are consistent.
Author Response
This manuscript has been improved after revision based on the comments from reviewers. However, there are still some minor mistakes in the manuscript. The authors need to read through the manuscript to correct these errors before publication. Some comments have been shown below.
Minor Comments:
- In the abstract section, the font size of ‘To collect various molecular data from the’ obviously is different from other words.
We have fixed it. Thank you
- In line 51, page 2: “In particular, considering that EVs can pass through the BBB, there are also attempt…”. BBB should not be abbreviated since it appeared in the manuscript for the first time.
We have fixed it. Thank you
- In line 179, page 5: “…RNAs such as miRNA/lncRNA etc.” There should be comma in front of ‘etc’.
We have fixed it. Thank you
- In lines 285-292, page 9, please make sure that all sizes and fonts are consistent.
We have fixed it. Thank you
